# Asymmetry of Endocast Surface Shape in Modern Humans Based on Diffeomorphic Surface Matching

Sungui Lin [1,2], Yuhao Zhao [1,2] and Song Xing [1,*]

1   Key Laboratory of Vertebrate Evolution and Human Origins, Institute of Vertebrate Paleontology and Paleoanthropology, Chinese Academy of Sciences, Beijing 100044, China; sglin@ivpp.ac.cn (S.L.); zhaoyuhao@ivpp.ac.cn (Y.Z.)
2   College of Earth and Planetary Sciences, University of Chinese Academy of Sciences, Beijing 100049, China
*   Correspondence: xingsong@ivpp.ac.cn

**Abstract:** Brain asymmetry is associated with handedness and cognitive function, and is also reflected in the shape of endocasts. However, comprehensive quantification of the asymmetry in endocast shapes is limited. Here, we quantify and visualize the variation of endocast asymmetry in modern humans using diffeomorphic surface matching. Our results show that two types of lobar fluctuating asymmetry contribute most to global asymmetry variation. A dominant pattern of local directional asymmetry is shared in the majority of the population: (1) the left occipital pole protrudes more than the right frontal pole in the left-occipital and right-frontal petalial asymmetry; (2) the left Broca's cap appears to be more globular and bulges laterally, anteriorly, and ventrally compared to the right side; and (3) the asymmetrical pattern of the parietal is complex and the posterior part of the right temporal lobes are more bulbous than the contralateral sides. This study confirms the validity of endocasts for obtaining valuable information on encephalic asymmetries and reveals a more complicated pattern of asymmetry of the cerebral lobes than previously reported. The endocast asymmetry pattern revealed here provides more shape information to explore the relationships between brain structure and function, to re-define the uniqueness of human brains related to other primates, and to trace the timing of the human asymmetry pattern within hominin lineages.

**Keywords:** cerebrum; cerebellum; petalia; shape asymmetry; diffeomorphic surface matching

## 1. Introduction

The structural and functional asymmetries of the human brain have been extensively studied in the fields of medicine and biology, especially as they relate to handedness and cognitive function [1–9]. The morphological characteristics of the brain surface are pressed into the inner cranium and may be visualized on endocasts, which are casts of the interior portion of a cranium [10,11]. Previous studies have demonstrated that the left and right hemispheres of human endocasts are usually asymmetrical in shape [12]. It is worth mentioning that asymmetries are divided into three categories: directional asymmetry, anti-symmetry, and fluctuating asymmetry [13,14]. The most concerning asymmetry pattern is directional asymmetry, characterized by a consistent directional bias within a population [13,15]. "Anti-symmetry" represents the reverse pattern (direction) of the population's directional asymmetry and is restricted to a small portion of the population. The third type of asymmetry is "fluctuating asymmetry", which means the asymmetry pattern of a character is diverse without a particular direction in a population [15].

A local impression on the internal surface of a skull, resulting from a protrusion of one brain hemisphere relative to the other, has been referred to as "petalia" and may be visible on endocasts [7]. Left occipito-petalias have been frequently associated with right fronto-petalias, whereas the directional asymmetries of parieto-petalias and temporo-petalias are inconsistent in different research [16,17]. Petalial asymmetries have been demonstrated to

exist in a wide variety of hominids [3,12,15,18,19]. The particular petalia asymmetry pattern with a right-frontal and left-occipital bias, commonly recognized in modern humans and fossil hominins, is considered to be associated with handedness [2,19–22]. Previous studies have also shown that petalial asymmetries differ between males and females, with males having slightly stronger right-frontal and left-occipital lateralization [1,2].

In many cases, protrusions of brain hemispheres are associated with lobar asymmetries. Previous investigations have revealed that the right frontal lobe and the left occipital lobe are frequently wider than the opposite hemisphere in modern humans and great apes [7,12,23]. Indeed, a prominent geometric distortion of the hemispheres, known as Yakovlevian anti-clockwise torque, is frequently observed in human brains and endocasts [7,24]. Specifically, the Yakovlevian anticlockwise torque includes the left-occipital and right-frontal petalias, with the left occipital lobe extending across the midline over the right and wider/larger right frontal and left occipital regions [22]. More recently, endocasts of the genus *Homo* have revealed that the right hemisphere often has a greater surface area than the left, while the right parieto-temporal lobe and the left occipital lobe have larger surface areas than their contralateral regions [4]. In addition, the asymmetries of the cerebellum and temporal lobe have also attracted attention, though relatively little is known about what their structural asymmetries may reflect [25].

The study of the brain lateralization associated with language was one of the most profound discoveries for neurobiology and linguistics. The leftward asymmetry of Broca's area, as a motor speech area, including the pars triangularis and pars opercularis of the inferior frontal gyrus, was first identified by Broca in 1861 [26]. Functional magnetic resonance imaging (fMRI) studies have provided more evidence for the specialization of the left hemisphere for language [5,27]. Broca's area is referred to as Broca's cap on an endocast, representing a "protrusion of the orbital portion of the inferior frontal gyrus" [28]. Since the emergence of genus *Homo*, it is generally accepted that the left Broca's cap is larger and more prominent than the right [29–31]. Recently, a landmark-based quantitative study of the asymmetry of the third frontal convolution in endocasts suggested that the left Broca's cap of modern humans, although smaller in size, is more globular and better defined than the right side [32]. Wernicke's area is responsible for language comprehension and mainly includes the posterior portion of the superior temporal gyrus and middle temporal gyrus, as well as the inferior parietal lobule, which includes the angular gyrus and supramarginal gyrus [24]. Due to the lack of homologous anatomical markers, Wernicke's area is not well defined on endocast surfaces and displays an unclear pattern of asymmetry [5,33–35]. Generally, though, the planum temporale (the main cortical area of Wernicke's area) is larger on the left hemisphere than on the right [36–38].

As mentioned above, most of the work on the asymmetry of endocasts has been focused on the degree of anterior or posterior protrusion, the lateralization of some regions associated with speech (e.g., Broca's cap), or the relative width and area of frontal and occipital lobes [39]. Moreover, previous research usually compared the human endocast with that of great apes or other primates to reveal a shared directional asymmetry pattern or a particular characteristic rather than conduct a global comparison of the entire brain surface [3,12,32,40]. The breadth of endocast asymmetry patterns within modern humans remains unclear.

Due to methodological limitations, it is difficult to comprehensively quantify the asymmetry of an endocast's entire surface. Previous studies typically relied on morphological descriptions, linear measurements, and geometric morphometrics [41–44]. The development of a landmark-free surface deformation method, diffeomorphic surface matching (DSM), provides interesting research opportunities for evaluating morpho-architectural variation on endocasts [45–47]. Compared with geometric morphometrics, DSM does not rely on the definitions of landmarks and semi-landmarks to capture the shape of the whole anatomical structure and can dynamically display the shape variation among different specimens [45–47]. Analytic results based on DSM indicate that endocasts of *Australopithecus africanus* (Sts 5 and Sts 60) display a more elongated frontal bec and a substantially less elevated parietal area,

different from those of genus *Homo* [46]. Visualizations of sulcal patterns have contributed more information to taxonomic identification in Old World monkeys [45]. Additionally, this deformation-based approach has been applied to dental materials and the vestibular apparatus [48,49]. However, no studies focusing on the morphological asymmetry of endocasts in modern humans have yet used this landmark-free method.

In the present study, 58 endocasts of archaeological modern Chinese crania were virtually reconstructed with high-resolution computed tomography and three-dimensional virtual technology. Landmark-free diffeomorphic surface matching analysis was performed to quantify and visualize the shape variation of the endocasts. We aim to quantify individual variation in the asymmetry of the endocast surface shape and analyze the variation of asymmetry patterns between the left and right hemispheres within the modern human population, as well as tentatively discuss the correlations between the structural and functional asymmetry of human brains.

## 2. Materials and Methods

### 2.1. Materials

In total, 58 adult endocasts, including 28 females and 30 males, were collected from the same archaeological site in Yunnan Province of Southwestern China, dated to about 300 years ago [50]. The skulls were well preserved or only minorly damaged in a way that would have no significant influence on the endocast reconstruction. The sexual assignment of specimens relied on diagnostic characteristics of the pelvis and cranium [50].

### 2.2. Endocast Reconstruction and Processing

All of the modern human specimens investigated in this study were scanned by a 450 KV industrial CT scanner with a spatial resolution of 160 μm (designed by the Institute of High Energy Physics, Chinese Academy of Sciences, and housed at the Key Laboratory of Vertebrate Evolution and Human Origins, Institute of Vertebrate Paleontology and Paleoanthropology, Chinese Academy of Sciences) [51]. The virtual reconstruction of each endocast was performed through semi-automatic threshold-based segmentation via the Mimics v. 17.0 software (Materialise, Leuven). A three-dimensional (3D) mesh of each endocast was generated and saved as an STL file in Mimics, and then imported into MeshLab software (Bangalore, India) [52] for 'Cleaning and Repairing' and a resulting 3D surface was obtained.

Considering the presence of Yakovlevian anticlockwise torque, the cerebral longitudinal fissure is not entirely on the midsagittal plane. Thus, the central axis or central plane separating a complete left hemisphere from its right side is difficult to determine. To investigate shape differences of the endocast between the right and left hemispheres, mirrored versions of each original specimen were created via Avizo v. 8.0 (FEI Visualization Sciences Group, Houston, TX, USA) [15,25]. The left side of the original endocast corresponds to the right side of the mirrored version and vice versa, as shown in Figure 1. Therefore, a total of 116 cases of endocast surfaces were included in the following deformation analyses. In this way, we could obtain a symmetrical mean shape by averaging the original and mirrored endocast of each individual in the following steps. The shape asymmetry of endocast surfaces was analyzed by calculating the deformation between each original endocast and its mirrored endocast.

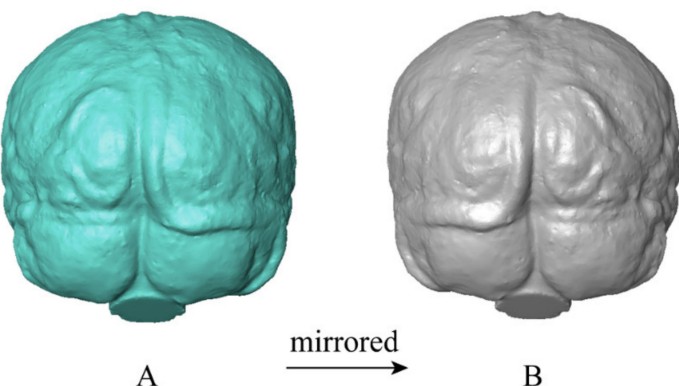

**Figure 1.** The original endocast (**A**) and the mirrored endocast (**B**) of the same individual in occipital view.

### 2.3. Diffeomorphic Surface Asymmetry

All surfaces were superimposed and aligned in Avizo through translation, rotation, and dilation (scaling) to eliminate differences, except for shapes. The aligned surfaces were exported as PLY files and then transformed to VTK format by ParaView v. 5.6.0 software (Kitware Inc., Clifton Park, NY, USA). The set of VTK data were imported into Deformetrica v. 4.3 (Paris, France) [53] to carry out the diffeomorphic calculation. The outputs include the spatial coordinates of control points defining the deformable space (9200 in this study), the momenta vectors recording the deformation information of each control point, and a symmetric endocast configuration representing the global mean shape [45,54].

For each endocast specimen, the vector difference at each control point was calculated by subtracting the momenta vector of the mirrored one from its counterpart of the origin one. Then, the surface asymmetry was quantified by an asymmetrical matrix depositing the vector differences at all control points.

A non-center principal component analysis (PCA) using the "RToolsForDeformetrica" [55] and "ade4" v. 1.7-17 [56] packages for R v. 4.0.4 [57] was carried out on an array storing the asymmetrical matrices of all specimens. The "ggplot2" package [58] was used to visualize the result of the PCA. A scatter plot of the second principal component (PC2) against the first principal component (PC1) displayed the distributional relationship of each specimen. The deformations displaying the asymmetric patterns at the four extremes were computed via Deformetrica and visualized in ParaView v. 5.6.0. These deformations were displayed in a form of colormap from dark blue (more constricted compared to the opposite) to red (more expanded compared to the opposite).

The mean matrix averaging the asymmetrical matrices of all specimens was calculated to exhibit the general pattern of surface asymmetry. The deformation for this mean matrix was also computed via Deformetrica and visualized in ParaView v. 5.6.0.

## 3. Results

### 3.1. Principal Component Analysis

The first two principal components account for 27.06% and 14.32% of the total shape asymmetry variation. The PC1 indicates that there is no clear directional trend for the two asymmetrical types represented by the shapes at the two extremes (Figure 2). At the positive extreme of PC1 (Figure 3A), the frontal, anterior parietal, and anterior temporal lobes in the right hemisphere are more bulged compared to the left hemisphere, whereas the occipital, posterior temporal, and posterior parietal lobes in the left hemisphere project more posteriorly and laterally than the right side. With increasing values along PC1, the left cerebellar lobe bulges and protrudes more posteriorly, while the right cerebellar lobe exhibits the opposite trend. Comparatively, the asymmetrical pattern in the negative-value end of PC1 shows a contrary trend to that at the positive end (Figure 3B).

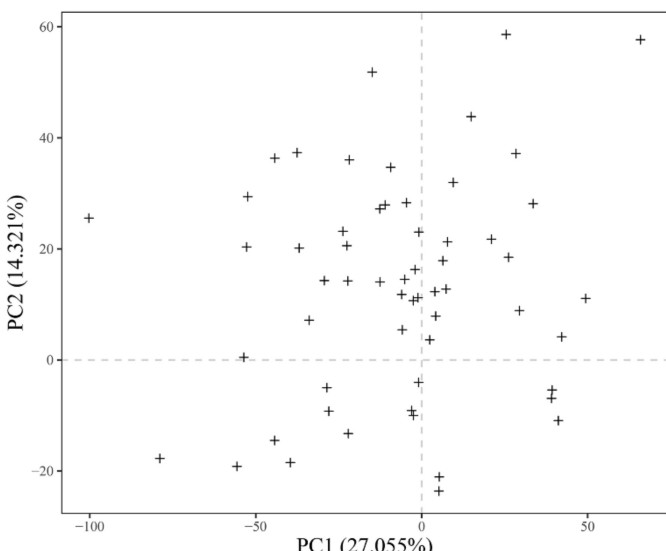

**Figure 2.** Bivariate plot of PC2 against PC1 based on diffeomorphic surface matching (DSM) analysis for endocast shape asymmetry variation.

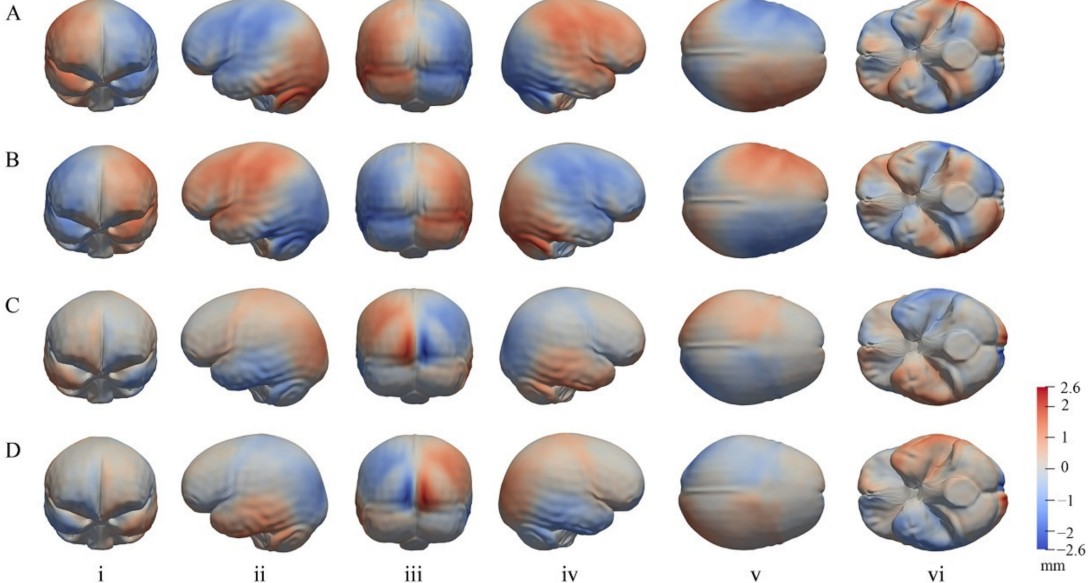

**Figure 3.** The virtual asymmetric shapes of endocast surfaces at the positive and negative extremes of PC1 (**A**,**B**) and PC2 (**C**,**D**) in frontal (**i**), left lateral (**ii**), occipital (**iii**), right lateral (**iv**), dorsal (**v**), and basal (**vi**) views. The colormap shows the degree of asymmetry in endocast surface shape. The dark blue and red represent the most constricted and expanded area of one hemisphere relative to the other half (in millimeter).

As shown in Figure 2, PC2 presents directional asymmetry: about 74% of the individuals are distributed in the positive-value half of PC2 and 26% in the negative-value half. The shape asymmetry pattern of the endocast at the dominant side of the distribution (positive extreme end of PC2, Figure 3C) is mainly shown as left-occipital and right-frontal petalial asymmetries. The superior and middle-frontal convolutions present a slight rightward asymmetry, while the inferior frontal convolution shows a slight leftward asymmetry. The temporal lobe presents a rightward asymmetry. The parietal-occipital lobe presents a leftward asymmetry. The cerebellum shows a double asymmetry with a leftward anterior lobe and rightward posterior lobe, but to a low degree. The negative extreme end of PC2

(Figure 3D) displays the reverse trend of the asymmetrical pattern, namely anti-symmetry of those at the positive extreme end of PC2.

### 3.2. The General Pattern of Endocast Surface Asymmetry

The mean asymmetric shape of the endocast surface, averaging the asymmetrical deformation of all individuals, is shown in Figure 4. It reveals a directional asymmetry pattern consistent with the positive-value end of PC2 but displays more detailed local asymmetries.

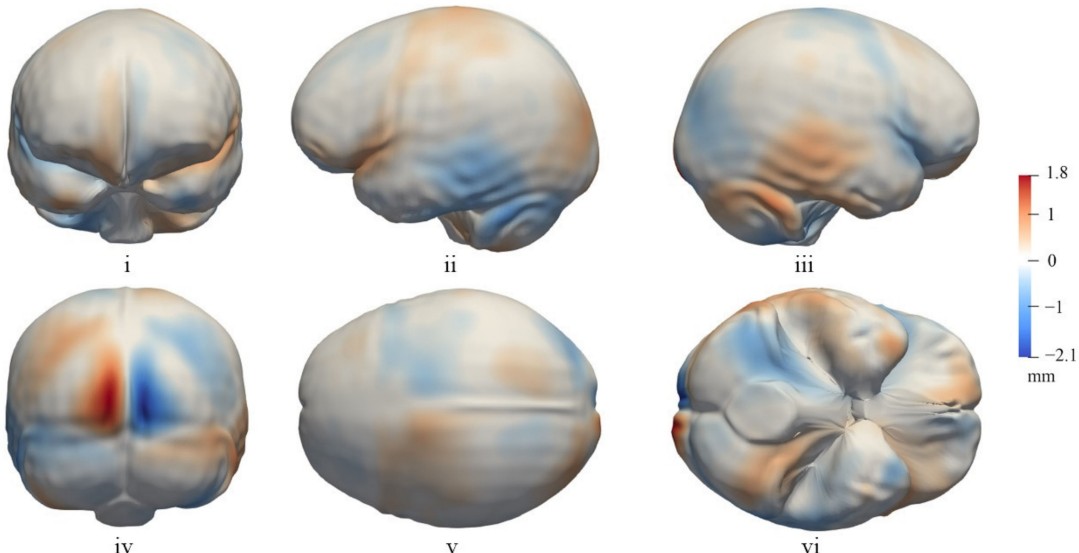

**Figure 4.** The mean asymmetric shape of the endocast surface in frontal (**i**), left lateral (**ii**), right lateral (**iii**), occipital (**iv**), dorsal (**v**), and basal (**vi**) views. The colormap shows the degree of asymmetry in endocast surface shape. The dark blue and red represent the most constricted and expanded area of one hemisphere relative to the other, and the displacement within 5% from symmetric shape is shown in white (in millimeter).

In the frontal view (Figure 4i), the frontal lobe has a strip adjacent to the longitudinal cerebral fissure, showing expansion on the right hemisphere and contraction on the left side. As a result, the right hemisphere protrudes more anteriorly than the opposite side and slightly bends the anterior interhemispheric fissure towards the left. The left inferior frontal convolution (Figure 4ii), an area that includes Broca's area, is extended more anteriorly, laterally, and ventrally relative to the right side, whereas the right inferior frontal convolution (Figure 4iii) appears more flattened and elongated antero-posteriorly. Additionally, the right frontal bec is more elongated than that of the opposite hemisphere and the ventral surface of the frontal lobes in the left hemisphere is more bulged than the right (Figure 4vi).

In terms of the local shape asymmetry of the temporal lobe (Figure 4ii,iii), the posterior portion, involving all three convolutions, is more inflated in the right hemisphere than the left. The right temporal lobe protrudes more inferiorly than the left one, resulting in a right temporal petalia (Figure 4vi). Additionally, the left temporal lobe appears to be shorter and the left Sylvian fissure is displaced anteriorly to a greater extent than the right. The parietal lobe also shows complex shape asymmetry (Figure 4v): the anterior portion adjacent to the frontal lobe and the posterior portion adjacent to the occipital lobe are more bulged on the left parietal lobe than the right, while an elliptic region, roughly corresponding to the superior parietal lobule, appears to protrude more on the right hemisphere than the left. Asymmetries in the temporal and parietal lobes suggest that the directional asymmetry in the surface of the region corresponding to Wernicke's area is in favor of the right side.

The greatest degree of local asymmetry in the endocast surface was observed in the occipital region (Figure 4iv): the portion of the left occipital lobe near the indentation of the

superior sagittal sinus protrudes significantly backward while the relative position of the right hemisphere contracts inward compared to the opposite side. Additionally, the left occipital lobe extends more medially than the right, bending the posterior interhemispheric fissure towards the right.

The surface shape of the cerebellum shows bilateral asymmetry (Figure 4iv,vi): the anterior portion of the cerebellum extends more anteriorly and ventrally in the left hemisphere than the right, whereas the posterior portion of the cerebellum extends more posteriorly and superiorly in the right hemisphere than the left. The cerebellum thus has the appearance of an anticlockwise twisting torque in basal view.

## 4. Discussion

### 4.1. Global Endocast Asymmetry

In this study, we performed surface matching using diffeomorphisms to quantify and visualize the asymmetry of human endocasts. We found that two types of lobar asymmetries categorize the majority of the global asymmetry variation. These two types of lobar asymmetries correspond to the clockwise and counterclockwise distortion of the global brain, with the cerebrum and cerebrum being consistent in the deformation trends. Previous studies revealed that the right frontal, right parieto-temporal, and the left occipital lobes have larger surface areas than the contralateral sides [4]. Measurement of lobe volumes using MRI found rightward asymmetries for the frontal and temporal lobes, and leftward asymmetries for the parietal and occipital lobes in right-handed twin pairs [20]. Here, we find that the rightward asymmetries of frontal, anterior parietal, and anterior temporal lobes, and leftward asymmetries of occipital, posterior temporal, and posterior parietal lobes have a roughly equivalent distribution compared to the reverse asymmetry pattern in this population. This indicates that the lobar asymmetry has a more complicated pattern of shape, surface area, and volume asymmetry.

### 4.2. Local Asymmetries of the Cerebrum

The shape at the positive-value extreme of PC2 and the deformation-based mean asymmetric shape both show a similar pattern of directional asymmetry. This directional asymmetry pattern is about three times as common as its anti-symmetry in the modern population. The general asymmetry pattern illustrated here is similar to the results of the previous study using geometric morphometrics to quantify hominid endocranial asymmetry [15] but reveals more details about the local asymmetries.

The petalia asymmetry pattern of endocasts in modern humans is typically characterized by the right frontal and left occipital lobes protruding outward more than the opposite side. Deformation results in the present study indicate that the leftward occipital petalia is much more prominent than the rightward frontal petalia, which is consistent with analyses of geometric morphometrics and linear measurements in endocasts of humans [15,59]. The temporal lobe presents a right petalia projecting inferiorly, which has not been observed in previous studies.

We found that the asymmetry in the posterior part of the temporal lobe favors the right side and the local asymmetries of the parietal lobe are complex depending on the mean asymmetric shape. In this context, the surface of Wernicke's area is primarily skewed to the right. Previous MRI research showed a complicated asymmetry in the temporal lobe. Kitchell and colleagues [5] have reported that the superior temporal sulcus is rightward-asymmetric while the planum temporale is leftward-asymmetric. A rightward asymmetry in the depth of the superior temporal sulcus ventral to Heschl's gyrus is known to be widely present in modern humans but rare in chimpanzees [60]. Due to the fact that the planum temporale and superior temporal sulcus are internal anatomical structures, it is difficult to define the contour of these area on an endocast. Therefore, the asymmetry of Wernicke's area discussed here is roughly based on the surface shape of the posterior parts of superior and middle-temporal convolutions.

A leftward asymmetry of Broca's area in the inferior frontal convolution has been identified as a feature commonly found in hominins and related to the brain lateralization associated with language [11,29]. Indeed, quantitative measurements revealed that the Broca's cap is larger in the right but more clearly defined in the left in modern humans [25,28,32]. In the present study, the inferior frontal convolution is more flattened and elongated antero-posteriorly in the right hemisphere, while the left inferior frontal convolution extends more laterally, anteriorly, and ventrally relative to the right side. As a result, the left Broca's cap appears to be more globular. In the quantitative study of *H. sapiens* endocast by Balzeau and colleagues [32], the left Broca's cap was also found to be more globular than the right side, but the length and the size of the third frontal convolution displayed a rightward asymmetry. This observation is supported and upheld by the results of the present study. Wada and colleagues [37] measured the visible cortical area on the frontal operculum (including both the pars opercularis and a posterior portion of the pars triangularis) and found that the left side was smaller than the right. However, Falzi and colleagues [61] measured the cortical surface area of Broca's area (including both the extra-sulcal and intra-sulcal cortex) and revealed that the left Broca's area was significantly larger than the right one. The difference in these results is due to the deeper fissure of the cortex in the left Broca's area [62] and perhaps leads to the larger size on the right but more globular shape on the left Broca's area.

### 4.3. Asymmetry of the Cerebellum

The cerebellum is responsible for controlling movement and coordinating balance, as well as for regulating cognition and emotion through information circuits with the non-motor cortex in the prefrontal and posterior parietal [63–65]. Previous studies have found a leftward asymmetry of the anterior cerebellum and a rightward asymmetry of the posterior cerebellum [25]. Here, the cerebellum shows a double asymmetry in which the right posterior cerebellar lobe extends more posteriorly and superiorly than the left, and across the midline, whereas the left anterior cerebellar lobe extends more anteriorly and ventrally. Therefore, the surface shape of the cerebellum appears as a twisting effect in the opposite direction relative to the Yakovlevian anticlockwise torque of the cerebrum. There is evidence that the motor and non-motor cortex in the left and right hemispheres of the cerebrum show strong preferential correlations with the related functional areas in the contralateral cerebellum [66–68]. In addition, the region and degree of functional lateralization in the cerebellum are correlated with that of lateralization in the cerebrum [6]. With that in mind, the cerebellum may possess roughly similar asymmetrical patterns of function and structure to the cerebrum [6].

### 5. Significance and Conclusions

Here, we quantified and visualized the asymmetry of endocast surface shapes in a modern human population using landmark-free DSM. Like previous studies, we have found the dominant asymmetry pattern to have left-occipital and right-frontal petalias, a more globular left Broca's area compared to the right, and a double asymmetry in the cerebellum. In addition, our results reveal more information of the asymmetry pattern in parietal and temporal lobes. Brain structural asymmetry is extensively involved in previous studies and often associated with functional lateralization. For example, the left hemisphere is generally dominant for language, with a more prominent Broca's area and a larger planum temporale [26,69]; besides, right-handed individuals often exhibit a more pronounced left-occipital and right-frontal petalias asymmetry than non-right-handed individuals [2]. Additionally, the cortical thickness asymmetry in a specific region of the postcentral gyrus correlated with hand preference, where right-handers dominated by the left hemisphere had a less rightward/more leftward shift of neural resources [9].

Our findings support previous MRI studies and confirm the validity of endocasts for obtaining valuable information on encephalic asymmetries [1,10,15,20,37,59]. Specifically, we find that the surface shape of the temporal language comprehension area (i.e., Wer-

nicke's area) presents a rightward asymmetry, which is different from the asymmetrical pattern of the motor speech area (i.e., Broca's area). Thus, the hemisphere dominance for language in terms of shape asymmetries, reflected by the endocast surface, might not be completely leftward. Furthermore, a rightward temporal petalia and a complex asymmetry pattern of the parietal lobe were also revealed. Whether these asymmetry features in the endocast surface can be correlated with a function of the brain needs to be investigated in combination with MRI studies as well as other morphological and functional studies in the future.

The evolution of the unique structure of the human brain has long been explored by analyzing the evolutionary sequence of endocasts and comparing humans and other primates [3,40,70,71]. A detailed understanding of the asymmetric patterns of the brain structure in modern humans is central to this topic. The PCA results presented in this study demonstrate that modern humans present a fluctuating asymmetry on the endocast surface, as represented by two balanced components of left parietal/right occipital lobes or right parietal/left occipital lobes. On the other hand, the non-center PCA results and the average asymmetric shapes show that most individuals exhibit a prevalent directional asymmetry, as discussed earlier in this paper. According to previous studies, the brain of great apes shows a similar right-frontal and left-occipital directional asymmetry in the width of lobes, and a similar but less variable and low-degree fluctuating asymmetry in components of the petalias [3,15,23]. Moreover, Balzeau and colleagues [32] have found that *Pan paniscus* shares a common pattern of asymmetries in the third frontal convolution with *Homo sapiens* through the quantitative study of endocasts. Great apes have also been shown to have leftward asymmetries in the size of the planum temporale and Broca's area [72,73]. Additionally, previous studies have revealed that levels of brain asymmetry varied in the evolution process of *Homo* species, especially when the common ancestor of *Homo heidelbergensis*, *Homo neanderthalensis*, and *Homo sapiens* emerged [40]. This study reveals that modern humans have a more complex pattern of endocast asymmetry than previously understood, which involves both fluctuating and directional asymmetry in different lobes. Considering this new understanding of endocast asymmetry, it is necessary to assess whether the asymmetry pattern in modern humans is also present in non-human primates and whether it is present in particular stages of hominin evolution.

**Author Contributions:** Conceptualization, S.X. and S.L.; methodology, Y.Z. and S.X.; software, S.L.; validation, S.L., Y.Z. and S.X.; formal analysis, S.L.; investigation, S.L.; resources, S.X.; data curation, S.L. and S.X.; writing—original draft preparation, S.L.; writing—review and editing, S.X.; visualization, S.L.; supervision, S.X.; project administration, S.X.; funding acquisition, S.X. All authors have read and agreed to the published version of the manuscript.

**Funding:** This work was supported by the Strategic Priority Research Program of the Chinese Academy of Sciences (no. XDB26000000) and the National Natural Science Foundation of China (41872030).

**Institutional Review Board Statement:** Not applicable.

**Informed Consent Statement:** Not applicable.

**Data Availability Statement:** The data supporting this study are available from the corresponding author on reasonable request.

**Acknowledgments:** The authors thank Yemao Hou and Jing Zuo for their help in CT scanning and 3D reconstruction processing. Xiujie Wu provided the Mimics files of 3D reconstruction. Mackie O'Hara-Ali helped us in revising the manuscript.

**Conflicts of Interest:** The authors declare no conflict of interest.

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
