# Peer review of "Asymmetry of Endocast Surface Shape in Modern Humans Based on Diffeomorphic Surface Matching"

_symmetry, doi:10.3390/sym14071459_

Round 1
Reviewer 1 Report
The manuscript “Asymmetry of endocast surface shape in modern human based on diffeomorphic surface matching” submitted for publication to MDPI Symmetry is appropriate for the readership of this journal but in its current state does not present any new insights other than presenting an application of a methodology without a research question/concept/idea/hypothesis/discussion. It has potential to be improved, I think, and I hope that my comments and questions can be of help in this process. Most importantly, the authors should discuss what the message of this manuscript is.
Unordered comments as they came up during reading the manuscript:
1) Please clarify if it is required to define a midsagittal plane for your methodology. It seems so from Figure 1 but if I understood your explanations in the text correctly, it is not required. This is important because the midsagittal plane can be asymmetric itself and largely influence the measurements of asymmetry depending on how it is defined.
2) Please elaborate about the study design. Why are we looking at original and mirror-imaged endocast data in one plot, is this saying anything about asymmetry? Why are females and males grouped? Is there an expectation that sexes are different in endocast asymmetry? What is the hypothesis of the paper? What do you want to explore with these data? Please elaborate and discuss.
3) Aside from recent critiques of between group principal component analyses (Bookstein, F.L. Pathologies of Between-Groups Principal Components Analysis in Geometric Morphometrics. Evol Biol 46, 271–302 (2019). Cardini, A., O’Higgins, P. & Rohlf, F.J. Seeing Distinct Groups Where There are None: Spurious Patterns from Between-Group PCA. Evol Biol 46, 303–316 (2019)), it is not clear to me why you use this ordination technique in the first place. bgPCA was intended to separate a priori groups from each other and to be still able to interpret within-group variation as compared to between group variation without a distortion of the morphospace. But original and mirror-imaged groups are not groups in terms of biological questions or hypotheses but mere technical constructs to measure asymmetry. And sexes as groups make sense but the manuscript does not introduce any ideas/concepts/hypotheses that and why differences in female and male endocast asymmetry should be investigated. I suggest using standard PCA of endocast shape.
4) Figure 2 shows variation in shape and that female and male endocast shape mostly overlap. And we can see that mirror-imaged endocasts are different but that is not really an analysis of asymmetry. I recommend thinking of another method like a PCA of asymmetry measures to get an impression about the variation in asymmetry, an interesting measure that is not interpretable from your figure now.
5) Please clarify what you mean with or how you define “to each extreme shape” in line 158. The individual with the most negative value on PC 1 and 2 catches my eye in the figure (the most asymmetric male), while I guess you are talking about the most negative value along the y-axis as visualized in Figure 3. You write about 4 deformations, but Figure 3 shows only one extreme (the only one that makes sense in my opinion). That’s a bit confusing. Furthermore, for more clarity, I suggest emphasizing in your text that the global mean shape (line 161) is a symmetric endocast, because of the mirror-imaged versions you included in the analysis.
6) In Figure 3, I recommend adjusting the colormap in a way that highlights interesting features and choose white instead of grey for symmetric regions for a better visualization.
7) I like the descriptions of (regional) asymmetries. This is very helpful.
Author Response
Responds to the reviewers' comments:
Reviewer #1
The manuscript “Asymmetry of endocast surface shape in modern human based on diffeomorphic surface matching” submitted for publication to MDPI Symmetry is appropriate for the readership of this journal but in its current state does not present any new insights other than presenting an application of a methodology without a research question/concept/idea/hypothesis/discussion. It has potential to be improved, I think, and I hope that my comments and questions can be of help in this process. Most importantly, the authors should discuss what the message of this manuscript is.
Response: Thanks for the positive evaluation of our work! We have rewritten the Abstract, Aim, and Discussion to elaborate the goal of this work and meaning of the results. In this work, we aim to quantify individual variation in asymmetry of endocast surface shape and analyze the variation of asymmetry patterns between the left and right hemispheres within the modern human population. This study confirms the validity of endocasts for obtaining valuable information on encephalic asymmetries and reveal a complicated asymmetry pattern of the human endocast in different regions of the cerebral lobes apart from some functional areas or petalias commonly involved in previous studies. The endocast asymmetry pattern revealed here provides more shape information to explore the relationships between brain structure and function, to re-define the uniqueness of human brains related to other primates, and to trace the timing of the human asymmetry pattern within hominin lineages.
1) Please clarify if it is required to define a midsagittal plane for your methodology. It seems so from Figure 1 but if I understood your explanations in the text correctly, it is not required. This is important because the midsagittal plane can be asymmetric itself and largely influence the measurements of asymmetry depending on how it is defined.
Response: Figure 1 was used to illustrate the process of mirroring and it has been revised in the new manuscript. It is not required to define a midsagittal plane for our methodology.
2) Please elaborate about the study design. Why are we looking at original and mirror-imaged endocast data in one plot, is this saying anything about asymmetry? Why are females and males grouped? Is there an expectation that sexes are different in endocast asymmetry? What is the hypothesis of the paper? What do you want to explore with these data? Please elaborate and discuss.
Response: The whole study has been re-designed and more details have been provided in the revision. Instead of using original and mirror-imaged endocast as the studying sample, we use the asymmetrical matrix (calculated by subtracting the momenta vector of the mirrored one from its counterpart of the origin one) in the analysis. Also, we have removed the bgPCA completely and use PCA.
In the PCA, we won’t separated Males from Females in order to focus on the individuals variations. Also, Females and Males generally overlap with each other along PC1 and PC2.
The paper does not include specific hypothesis. Considering that most of previous studies for the asymmetry of endocast have been focusing on the petalias asymmetry and other specific directional asymmetry using GM and linear measurement, our original goal was to more comprehensively explore the endocast asymmetry patterns of modern human in more detail with a new approach based on diffeomorphism and to show the shape deformation dynamically.
In this work, we aim to quantify individual variation in asymmetry of endocast surface shape and analyze the variation of asymmetry patterns between the left and right hemispheres within the modern human population. The endocast asymmetry pattern revealed here could provide more shape information to explore the relationships between brain structure and function, to re-define the uniqueness of human brains related to other primates, and to trace the timing of the human asymmetry pattern within hominin lineages in the future work.
3) Aside from recent critiques of between group principal component analyses (Bookstein, F.L. Pathologies of Between-Groups Principal Components Analysis in Geometric Morphometrics. Evol Biol 46, 271–302 (2019). Cardini, A., O’Higgins, P. & Rohlf, F.J. Seeing Distinct Groups Where There are None: Spurious Patterns from Between-Group PCA. Evol Biol 46, 303–316 (2019)), it is not clear to me why you use this ordination technique in the first place. bgPCA was intended to separate a priori groups from each other and to be still able to interpret within-group variation as compared to between group variation without a distortion of the morphospace. But original and mirror-imaged groups are not groups in terms of biological questions or hypotheses but mere technical constructs to measure asymmetry. And sexes as groups make sense but the manuscript does not introduce any ideas/concepts/hypotheses that and why differences in female and male endocast asymmetry should be investigated. I suggest using standard PCA of endocast shape.
Response: we have removed the bgPCA completely and used PCA. The female and males are not separated in the PCA, as female and male groups generally overlap with each other for the whole scope along PC1 an PC2.
4) Figure 2 shows variation in shape and that female and male endocast shape mostly overlap. And we can see that mirror-imaged endocasts are different but that is not really an analysis of asymmetry. I recommend thinking of another method like a PCA of asymmetry measures to get an impression about the variation in asymmetry, an interesting measure that is not interpretable from your figure now.
Response: We have removed the bgPCA completely and used PCA.
5) Please clarify what you mean with or how you define “to each extreme shape” in line 158. The individual with the most negative value on PC 1 and 2 catches my eye in the figure (the most asymmetric male), while I guess you are talking about the most negative value along the y-axis as visualized in Figure 3. You write about 4 deformations, but Figure 3 shows only one extreme (the only one that makes sense in my opinion). That’s a bit confusing. Furthermore, for more clarity, I suggest emphasizing in your text that the global mean shape (line 161) is a symmetric endocast, because of the mirror-imaged versions you included in the analysis.
Response: In this study, non-linear deformations between different endocast surfaces were calculated via the Deformetrica software and it can display the trend of shape changes from one object to another. The extreme shape of endocast in one principal component axis does not represent a specific individual, but rather a virtual shape by computational synthesis, as an extreme case resulting from statistical analysis in shape deformation along an axis of principal components. Meanwhile, the trend from the average shape to the extreme shape is shown by a form of colormap. The four deformation trends of the first two principal components of bgPCA are shown in the Figure 1. Since the original manuscript focuses on the shape asymmetry pattern of the original, we only showed the results of the negative extreme of bgPC2. In the revised manuscript using standard PCA method to directly analyze the shape asymmetry deformation, we have shown the shapes of the four extremes of the first two principal components (Figure 2, i.e., Figure 3 in the revised manuscript). The corresponding modifications have been reflected in the revised manuscript.
6) In Figure 3, I recommend adjusting the colormap in a way that highlights interesting features and choose white instead of grey for symmetric regions for a better visualization.
Response: Thanks for the suggestions. Considering that in the four deformations from principal component analysis, the current colormap form shows a better gradient effect, so it has not been modified. The mean asymmetric shape of the endocast surface averaging the asymmetrical deformation of all individuals is exhibited in Figure 3 (i.e., Figure 4 in the revised manuscript), using an adjusted colormap in which the displacement within 5% from symmetric shape is shown in white. It represents the most general pattern of endocast shape asymmetry in the population.
7) I like the descriptions of (regional) asymmetries. This is very helpful.
Response: Thanks for the support!
Reviewer 2 Report
The study by Sungui Lin, Yuhao Zhao, and Song Xing aims to quantify and visualize brain endocast asymmetry in modern humans using diffeomorphic surface matching analysis.
Although the results obtained are not particularly novel when compared with those obtained with MRI, they do confirm the validity of endocasts for obtaining valuable information on encephalic asymmetry.
In this regard, it would be interesting for the authors to make a general comment on the significance and validity of the results obtained, both at the end of the abstract and in the conclusions.
Minor issues:
Line 85: "However, relatively few studies focusing on the morphological variation of endocasts in modern humans have yet used this landmark-free method."
Which studies have used the DSM method? They should be referenced.
Citations 1-7 dealing with human brain asymmetries only cover the period up to 2013. It would be interesting to add to them a more recent quote, for example:
Kuo, F., & Massoud, T. F. (2022). Structural asymmetries in normal brain anatomy: A brief overview. Annals of anatomy = Anatomischer Anzeiger : official organ of the Anatomische Gesellschaft, 241, 151894. https://doi.org/10.1016/j.aanat.2022.151894
Likewise, the quotes dedicated to the relationship between cerebral laterality and handedness could be enriched with a more recent quote:
Sha, Z., Pepe, A., Schijven, D., Carrión-Castillo, A., Roe, J. M., Westerhausen, R., Joliot, M., Fisher, S. E., Crivello, F., & Francks, C. (2021). Handedness and its genetic influences are associated with structural asymmetries of the cerebral cortex in 31,864 individuals. Proceedings of the National Academy of Sciences of the United States of America, 118(47), e2113095118. https://doi.org/10.1073/pnas.2113095118
Figure 3 is not indicated in the text. Moreover, it would be interesting to have a reference in the text to the different views. For example for 3i and 3ii
Line 169-177
In frontal view of the frontal lobe (Fig. 3i), there is a strip of local area adjacent to the longitudinal cerebral fissure showing expansion on the right hemisphere and contraction on the left side, while the rest of the areas have the opposite trend. As a result, the right hemisphere protrudes more anteriorly than the opposite side, and slightly bends the anterior interhemispheric fissure towards the left. Additionally, the right frontal bec is more elongated than that of the opposite hemisphere and the ventral surface of the frontal lobes in left hemisphere is more bulged than the right side. Th lateral part of the frontal lobe (Fig. 3ii), an area that includes Broca’s area, is extended more anteriorly, laterally, and ventrally relative to the global mean shape in the left side of the endocast, whereas the right inferior frontal convolution appears more flattened and reduced.
This type of nformation would be helpful for the rest of the results section.
Author Response
Responds to the reviewers' comments:
Reviewer #2
The study by Sungui Lin, Yuhao Zhao, and Song Xing aims to quantify and visualize brain endocast asymmetry in modern humans using diffeomorphic surface matching analysis. Although the results obtained are not particularly novel when compared with those obtained with MRI, they do confirm the validity of endocasts for obtaining valuable information on encephalic asymmetry. In this regard, it would be interesting for the authors to make a general comment on the significance and validity of the results obtained, both at the end of the abstract and in the conclusions.
Response: Thanks for your support! We have added relevant information in the Abstract and end of the Discussions for significance and conclusion in the revised manuscript.
Minor issues:
1) Line 85: "However, relatively few studies focusing on the morphological variation of endocasts in modern humans have yet used this landmark-free method." Which studies have used the DSM method? They should be referenced.
Response: It has been revised in the manuscript.
2) Citations 1-7 dealing with human brain asymmetries only cover the period up to 2013. It would be interesting to add to them a more recent quote, for example: Kuo, F., & Massoud, T. F. (2022). Structural asymmetries in normal brain anatomy: A brief overview. Annals of anatomy = Anatomischer Anzeiger: official organ of the Anatomische Gesellschaft, 241, 151894. https://doi.org/10.1016/j.aanat.2022.151894. Likewise, the quotes dedicated to the relationship between cerebral laterality and handedness could be enriched with a more recent quote: Sha, Z., Pepe, A., Schijven, D., Carrión-Castillo, A., Roe, J. M., Westerhausen, R., Joliot, M., Fisher, S. E., Crivello, F., & Francks, C. (2021). Handedness and its genetic influences are associated with structural asymmetries of the cerebral cortex in 31,864 individuals. Proceedings of the National Academy of Sciences of the United States of America, 118(47), e2113095118. https://doi.org/10.1073/pnas.2113095118
Response: Thanks for providing these references, they are very helpful. We have added their findings that are relevant to our studies into the revision.
3) Figure 3 is not indicated in the text. Moreover, it would be interesting to have a reference in the text to the different views. For example for 3i and 3ii, Line 169-177, In frontal view of the frontal lobe (Fig. 3i), there is a strip of local area adjacent to the longitudinal cerebral fissure showing expansion on the right hemisphere and contraction on the left side, while the rest of the areas have the opposite trend. As a result, the right hemisphere protrudes more anteriorly than the opposite side, and slightly bends the anterior interhemispheric fissure towards the left. Additionally, the right frontal bec is more elongated than that of the opposite hemisphere and the ventral surface of the frontal lobes in left hemisphere is more bulged than the right side. Th lateral part of the frontal lobe (Fig. 3ii), an area that includes Broca’s area, is extended more anteriorly, laterally, and ventrally relative to the global mean shape in the left side of the endocast, whereas the right inferior frontal convolution appears more flattened and reduced. This type of information would be helpful for the rest of the results section.
Response: Thanks for the comments and suggestions! We have modified the contents.